

# Performance of open-path GasFinder3 devices for CH$_4$ concentration measurements close to ambient levels

Christoph Häni[1], Marcel Bühler[1,2,3], Albrecht Neftel[4], Christof Ammann[5], Thomas Kupper[1]

[1]School of Agricultural, Forest and Food Sciences HAFL, Bern University of Applied Sciences, Zollikofen, 3052, Switzerland
[2]Oeschger Centre for Climate Change Research, University of Bern, Bern, 3012, Switzerland
[3]Institute of Geography, University of Bern, Bern, 3012, Switzerland
[4]Neftel Research Expertise, Wohlen b. Bern, 3033, Switzerland
[5]Climate and Agriculture Group, Agroscope, Zürich, 8046, Switzerland

*Correspondence to*: Christoph Häni (christoph.haeni@bfh.ch)

**Abstract.** Open-path measurements of methane (CH$_4$) with the use of GasFinder systems has been frequently used for emission estimation with the inverse dispersion method (IDM), specifically from agricultural sources. It is common to many IDM applications that the concentration enhancement related to agricultural CH$_4$ sources is small, typically between 0.05 and 0.5 ppm, and accurate measurements of CH$_4$ concentrations are needed at concentrations close to ambient levels. The GasFinder3-OP (GF3) device for open-path CH$_4$ measurements is the latest version of the commercial GasFinder systems by Boreal Laser Inc. We investigated the uncertainty of six GF3 devices from side by side intercomparison measurements and comparisons to a closed-path quantum cascade laser device. Relative biases as high as 8.3 % were found and a precision between 2.1 and 10.6 ppm-m was estimated. These results deviate from the respective manufacturer specifications of 2 % and 0.5 ppm-m. Intercalibration of the GF3 devices by linear regression to remove measurement bias was shown to be of limited value due to drifts and step changes in the recorded GF3 concentrations.

## 1 Introduction

The experimental determination of methane (CH$_4$) emission rates from agricultural sources is a key element for emission inventories and for the developments of mitigation strategies. A large diversity of approaches to derive emission rates from measurements is available. Focusing on micrometeorological methods, they can broadly be divided into flux-based and concentration-based approaches. The latter combine measurements of the concentration enhancement downwind or above the source with the modelling of the dispersion of the concentration released by the source. One frequently applied concentration-based approach is the inverse dispersion method (IDM; Flesch et al., 2005) where, generally, two concentration measurements are used in parallel, placed up- and downwind of the source under investigation. It is in common to many IDM applications that the concentration enhancement related to agricultural CH$_4$ sources is small, typically between 0.05 and 0.5 ppm.



In recent years, open-path instruments became commercially available that determine the path-integrated concentration over measurement path lengths of up to several 100 meters. They are based e.g. on the determination of the absorption over a small wavelength range e.g. in the infrared spectrum (tunable diode laser technique for $CH_4$; DeBruyn et al., 2020). Regarding the IDM, path-integrated concentration measurements are preferable over point measurements, since they capture

a larger fraction of the emission related plume and, therefore, are less sensitive to variation and uncertainty in the measured wind direction.

On the other hand, it is more difficult to assess and control the quality of measurements by open-path gas analyzers in comparison to closed-path instruments. The latter can be checked or recalibrated periodically during a field campaign using common cylinder standards (also for multiple spatially separated instruments). This is usually not possible for open-path

devices with longer measurement paths. The use of cylinder standard gases is feasible for very short path lengths (few meters) but the corresponding calibration may not be representative for other setups with longer path lengths (DeBruyn et al., 2020). Therefore, the quality of open-path measurements in the field with path lengths of 10-100 m (or longer) needs to be tested in other ways using e.g. instrument internal quality indicators, plausibility checks, and intercomparisons of two or more instruments.

In this paper, we focus on the GasFinder3-OP (GF3) system for $CH_4$ measurements (Boral Laser Inc, Edmonton Canada) with the 'Lo-Range' calibration option. This open-path system has a very user-friendly design and is in the lower cost range of available instruments. It is an improved version of the GasFinder2 system, which has been frequently used to measure emission rates with the IDM (Flesch et al., 2007; Harper et al., 2010; McGinn et al., 2019; VanderZaag et al., 2014). The aim of this study is to characterize the stability and accuracy of the GF3 instruments for $CH_4$ measurements close to ambient

levels. We present an overview of several field campaigns including (i) an intercomparison between a GF3 device and a fast-response quantum cascade laser spectrometer (QCL) considered as a state-of-the-art reference and (ii) direct intercomparisons between various GF3 instruments. They served to generate a basis to correct the measurement data of individual GF3 instruments placed up- and downwind of emitting sources, which induced a low concentration enhancement where instrument stability and accuracy are particularly important. This article is written from the point of view of a GF3

instrument's end user.

## 2 Materials and Methods

### 2.1 GasFinder3-OP Instrument

The GF3 instrument from Boreal Laser Inc. is an open-path instrument with a tunable laser diode emitting in the infrared centered around 1654 nm where $CH_4$ shows a distinct absorption line. The measurement output of the GF3 is provided as

path-integrated concentration $C_{PI}$ in units of ppm-m that reflects the concentration integrated over the single path length (distance between laser source and reflector). The output data in units of ppm-m was converted to the path-averaged concentration C in units of ppm (i.e. divided by the single path length) and corrected with temperature and pressure





correction functions provided by the manufacturer. Six different open-path GF3 devices were used in this study (Table 1). The two devices OP-Ext and OP-1, as well as OP-3 and OP-5, had identical pressure and temperatures correction functions.

The 'Lo-Range' version of the GF3 for $CH_4$ measures in the range of 2 to 8500 ppm-m with a sensitivity (precision) of 0.5 ppm-m at a sample rate of 1 to 1/3 Hz as stated by the manufacturer (Boreal Laser Inc., 2020). The accuracy of the GF3 system is specified as 2 % of the reading (Boreal Laser Inc., 2018a) with a lower value for the 'typical accuracy' of 0.5 % of the reading (Boreal Laser Inc., 2018b). Details on the instrument are given in DeBruyn et al. (2020).

According to the manufacturer, a valid concentration measurement can be expected if the 'received power' of the reflected incoming laser beam is in the range of 50 to 3000 µW and if the goodness of fit between the sample and the calibration waveform quantified as R2 is above 0.85 (Boreal Laser Inc., 2018b). We decided to be stricter and kept data for further analysis only if the received power was in the range of 100 to 2500 µW (as suggested in Boreal Laser Inc., 2016) and R2 was equal or greater than 0.98. The quality-assessed data were aggregated to 1-minute and 30-minute average concentrations. Only averages resulting from a data coverage of 90 % or more of the respective time interval were retained for further evaluation.

## 2.2 Intercomparison Campaigns

In total, eight intercomparison campaigns were conducted at different sites in Switzerland with varying ranges of ambient concentrations of $CH_4$ (Table 2). Two campaigns, P16 and P17, with a focus on the comparison between GF3 devices and a QCL (QC-TILDAS, Aerodyne Research Inc.) as a reference system were conducted in Posieux (46°46'4.22"N / 7° 6'27.65"E) close to an animal housing facility (approx. 100 m north). The QCL is a closed-path instrument with a 20 m inlet tube flushed by a vacuum pump at 13 sL min[-1]. The sample air is analyzed in a multi-pass cell (0.5 L) with a fixed optical path length of 76 m. The cell is kept at constant temperature (294 K) and pressure (31 Torr). Due to the stabilized operation, the instrument exhibits a high precision (1 s) around 0.004 ppm or 0.2 % (Nelson et al., 2004; Wang et al., 2020).

Seven intercomparison campaigns including various GF3 instruments placed side by side were carried out at the following locations: A18 in Aadorf (47°29'19.03"N / 8°55'8.83"E) next to a dairy housing, K19 in Kaufdorf (46°50'34.60"N / 7°30'12.23"E), H19-1, H19-2 and H19-3 in Hindelbank (46°59'11.86"N / 7°28'22.01"E) close to a wastewater treatment plant, I19 in Ittigen (46°59'13.04"N / 7°28'20.38"E) in the vicinity of a biogas plant and P17 where both, the intercomparison of the GF3 as well as the comparison to the QCL was assessed. Different types of reflectors for the open-path instruments were in usage[1]. In the campaigns P16, P17 and A18, the 7-corner cube array type (suitable for path lengths between 45 and 75 m; Boreal Laser Inc., 2018a) was used, in H19-1, H19-2, H19-3 and I19, the 12-corner cube array type (suitable for path lengths between 75 and 200 m) was used, and in K19 both types were used.

---

[1] In 2016, when the first devices of GF3 (OP-1 to OP-3) were ordered, Boreal Laser Inc. recommended 7-corner cube array type reflectors for path lengths up to 200 m. Meshes of different grid sizes could be installed in front of the corner cubes for path lengths that are shorter than the specified range. Prior to the second order in 2019 (devices OP-4 and OP-5), the recommendation was adapted to use the 12-corner cube array type reflectors for path lengths up to 200 m.


During side-by-side intercomparisons, the laser beams of the GF3 devices were always aligned in parallel with small lateral distances of 1 to 2 m. Instrument and laser beam heights were between 1.3 and 1.7 m above ground. For the comparison to the QCL measurements, the QCL inlet was located approx. 4 to 12 m from the center of the laser beams 1.9 m above ground.

For the temperature and pressure correction of the GF3 instruments (Sect. 2.1) during the field campaigns, the temperature and pressure data from a close-by weather station was used. In A18, the weather station was situated 1.2 km away with a negligible difference in the elevation of approx. 6 m. At all other sites, the weather station was within 100 m of the devices. All measurements were conducted continuously, i.e. during day and night, in regions characterized by agricultural activities related to livestock production.

## 2.3 Data Evaluation

For a valid concentration comparison between the parallel instruments, the internal clocks of the individual devices were adjusted such that all concentration data were synchronous. This time synchronization was done by maximizing the covariance of the high-frequency concentration data in ppm between the individual instruments. For each day, the data was broken down to 1 second data (i.e. inserting repetition values where necessary) and the time shift with the highest covariance

was assessed. From these daily estimates of time shifts a constant time lag was estimated and corrected for each device and each campaign individually. Time lags around 2 to 5 s per day between the devices have been observed and corrected for.

In two intercomparison campaigns (P16 and P17) four different GF3 devices (OP-Ext, OP-1, OP-2 and OP-3) were compared to the closed-path point measurements by the QCL instrument based on the 30-minute averaged concentrations.

In seven intercomparisons (P17, A18, K19, I19, H19-1, H19-2 and H19-3), the GF3 devices OP-1, OP-2, OP-3, OP-4 and

OP-5 were compared by parallel measurements. The analysis of these intercomparisons are based on both, 1-minute and 30-minute averaged concentration data. The device OP-1 was running during all side-by-side campaigns and, thus, was selected as the (relative) reference instrument, i.e. any comparison was done with reference to OP-1.

Based on the synchronized time series, the concentration difference ΔC between the parallel instruments was calculated for each averaging interval. For characterizing the difference between devices, the median ΔC and the 'median absolute

deviation' (MAD) of ΔC over each campaign were determined for each pair of devices. The two quantities are robust estimates of the mean and variability of ΔC that are insensitive to outliers and do not rely on prescribed data distributions. For the ideal case of a Gaussian distribution, the MAD can be related to twice the standard deviation (comprising 95 % of the data) by multiplication with a factor of 2.9. The resulting value represents an estimate for the (random) precision of ΔC, whereas the median ΔC represents the (systematic) bias between the two instruments. The estimates of bias and precision

can be partitioned equally to both intercompared devices by dividing by the square root of 2 (according to Gaussian error propagation). Thus, the relative bias and the precision of an individual GF3 device for a campaign period was estimated as:

$$\text{rel. bias} = \frac{\text{median}(\Delta C)}{c_{\text{avg}}\sqrt{2}}, \tag{1}$$





$$\text{precision} = \frac{2.9 \times \text{MAD}(\Delta C)}{l_{\text{path}}\sqrt{2}}, \tag{2}$$

where the relative bias was expressed relative to the concentration average of the two devices $C_{\text{avg}}$ and the precision was

converted back to path-integrated concentrations $C_{\text{PI}}$ using the single path length $l_{\text{path}}$ of the GF3 device (in the case of the intercomparison of two GF3 devices the path lengths were averaged).

In addition to the concentration differences, the parallel measurements were also analyzed concerning their linear relationship using Deming regression that considers measurement errors from both instruments. The GF3 devices were analyzed with reference to OP-1. Coefficients from the linear regression and the predicted $\Delta C$ at OP-1 concentration levels

of 2 ppm and 4 ppm were reported for each device (OP-2, OP-3, OP-4 and OP-5) and campaign, if the number of observations exceeded 20 and the concentration range was large enough (difference between 0.025 and 0.975 quantiles greater than 0.4 ppm).

## 3 Results and Discussion

### 3.1 Intercomparison between GF3 and QCL

During the two intercomparison campaigns P16 and P17, the magnitude and temporal course of the GF3 concentrations measured by the devices OP-Ext, OP-1, OP-2 and OP-3 compared well to the concentration measured by the QCL, specifically for high frequency structures. Figure 1 shows 1.5 days of parallel QCL and OP-Ext measurement in campaign P16. However, when focusing on the lower end 'baseline' concentrations near 2.2 ppm the OP-Ext signal shows drifts and steps relative to the more stable QCL signal in the order of 0.2 ppm (shaded phases in Fig. 1). This corresponds to instrument

related changes in the path-integrated concentration of about 7.4 ppm-m (path length of 37 m).

At the 26 hours timestamp, a drift occurred dropping the concentration of OP-Ext from roughly 0.2 ppm above, to roughly 0.1 ppm below the QCL concentration. There is no indication of a deterioration of the measurement quality of the GF3 values during this period. The received laser beam power was always above 100 µW and the $R^2$ value for the waveform fit was greater than 0.98 (Sect. 2.1). Further, there was no correlation of the drift with the local weather data (air temperature,

wind direction, wind speed, relative humidity etc.; data not shown). The same applies to step changes and drifts of GF3 devices, typically over several hours, during other phases of the intercomparison campaigns. In some selected cases, step changes in the concentration could occur when there was activity related to device handling during operation (such as downloading data, checking the reference cell state etc.), as observed at hour 46 in Fig. 1. However, such device handling should not affect the measurements and it remains unclear, what exactly causes the signal changes. Since these drifts and

step changes cannot be distinguished from real changes in the ambient concentration without the information from a further parallel measurement, they affect the uncertainty in the GF3 measurements.

Bias and precision of the GF3 devices (Sect. 2.3) were estimated and compared to the accuracy (2 % of reading) and sensitivity (0.5 ppm-m) specified in the GF3 operation manual. The magnitude of the relative bias of the GF3 is higher than



the stated 2 %, with values ranging from -2.7 % to -8.3 % (Table 3). The $C_{PI}$ precision for the GF3 devices was determined
to 2.1 up to 10.6 ppm-m, which is between 4 and 21 times higher than the specified sensitivity of 0.5 ppm-m.

## 3.2 GF3 side by side intercomparisons

A cumulated dataset of totally 60 days with GF3 side-by-side measurements that passed the enhanced quality checks was
produced within the seven intercomparison campaigns P17, A18, K19, I19, H19-1, H19-2 and H19-3. It contains the periods,
during which at least two devices were running in parallel, i.e. the reference device OP-1 and at least one further instrument
(OP-2, OP-3, OP-4 or OP-5). Data from device OP-4 measured during the campaign H19-3 passed the quality check but has
been omitted in the further analysis due to an obvious jump in concentration (Fig. 2 and Fig. 3). The overall average $CH_4$
concentration was 2.14 ppm. One-minute averages ranged between 1.3 and 40.3 ppm with most of the data centered around
2 ppm.

Extended periods of $CH_4$ concentrations constantly below 1.88 ppm, the minimum of the monthly average background
165    concentration in Switzerland since 2016 (BAFU, 2019), could be observed with devices OP-1, OP-2 and OP-3. Overall,
shares of measured $CH_4$ concentration (1-minute averages) below 1.88 ppm were ranging from 0 % (OP-5) and 13 % (OP-4)
to 27 % (OP-2), 35 % (OP-3) and 41 % (OP-1), whereas values above 3.5 ppm rarely occurred 1 % (OP-2), 2 % (OP-1) and
3 % (OP-3, OP-4 and OP-5). This agrees with the systematically lower concentrations measured with the GF3 devices
compared to the measurement by the QCL device in the previous section.

Figure 3 shows the 30-minute averages of the recorded OP-1 concentration with the corresponding differences between the
measured concentration by the individual devices and the OP-1 concentration. The differences are generally small, but larger
deviations, as e.g. during the A18 campaign, occur.

Table 4 provides statistics on the differences between the GF3 devices OP-2 to OP-5 and the reference device OP-1
regarding directly comparable 30-minute concentration averages. The differences were determined in units of ppm and
transformed to ppm-m related to the path length of the GF3 device that has been compared to OP-1. The relative bias ranged
from -1.7 % to 8.0 % and the precision of $C_{PI}$ between 2.6 and 8.8 ppm-m, which lies within the range of the precision
estimates in Sect. 3.1. A large offset in the concentration, reflected by the relative bias, could be observed for OP-4 and OP-5
compared to concentration measurements from OP-1 (on average > 0.15 ppm higher). Devices OP-4 and OP-5 were acquired
two years later than instruments OP-1 to OP-3 and this offset may be due to a difference in the internal calibration by the
manufacturer between the instruments acquired in 2017 and the instruments acquired in 2019.

The devices OP-1 and OP-3 episodically showed dents in the concentration output that are in line with step decreases in the
received power. Figure 4 shows an example of such a dent recorded by OP-1 with OP-3 that measured in parallel as a
reference. The rapid loss of receiving power at 27.1 hours after device start seems to trigger a gradual loss up to 0.15 ppm in
the concentration of OP-1. A few minutes later a step change in the concentration by almost 0.2 ppm occurs, while the
received power is still low. We do assign these concentration variations to wrong concentration determination of the OP-1 as
the OP-3 concentration stays constant at the ambient background value slightly above 1.8 ppm. This indicates that a constant





threshold for the received power (50 or 100 μW) may be not sufficient for quality filtering. We noticed that the 'optimal' threshold varied up to 400 μW between individual instruments and campaigns.

Frequently, linear regression is used to correct for differences between instruments. There are two problems, however, that
can occur with this correction method for GF3 devices in the case of $CH_4$ concentration measurements close to ambient level. One problem arises if the dataset contains drifts and steps as shown in Fig. 1 and Fig. 4. Inspecting the A18 intercomparison between OP-1 and OP-2 closer (intercept: -0.04, slope: 1.04), a period of approximately 5.4 continuous days is apparent (around intervals 550 to 750 in Fig. 3) where OP-2 (and OP-3) recorded systematically higher concentrations than OP-1. If we separate this 'offset' period from the remaining part of the campaign (Fig. 5), we see that the regression
results are systematically different. The 'offset' period shows an intercept of 0.04 and a slope of 1.05, whereas we get an almost perfect 1:1 relationship for the residual time (intercept: 0.01, slope: 1.00). Using the overall regression results for the entire period (Table 5) instead of two separate periods thus introduces a bias in the evaluation.

The second problem is the observed, rather large variation in the intercalibration from one campaign to another (Table 5). Such a variation between different campaigns was also observed with GF3 devices for ammonia measurements by Baldé et
al. (2019). Concentration response of the instrument do change between different campaigns as seen by the regressions and can thus not be generalized. A significant difference in the predicted concentration between different campaigns can be seen for devices OP-2 and OP-5, e.g. within the same year 2019 (campaigns I19 and H19-3), intercalibrating OP-2 with OP-1 would provide significantly different 30 minute concentration estimates at concentration levels of 2 ppm and 4 ppm. Even though, in theory, an intercalibration of the devices after an IDM measurement campaign could solve the issue of differences
in the measurements, the necessary change in the setup to perform such an intercalibration could lead to a change in the response of the devices and the intercalibration would then be useless.

## 4 Conclusion

We found that the uncertainty in the measurements of several GasFinder3-OP instruments is higher than given in the specification provided by the manufacturer when measuring concentrations close to ambient levels. From on-site
intercomparisons at various field sites (side by side intercomparisons and comparisons to a reference QCL instrument), we estimate a bias up to 8.3 % of the reading and a precision between 2.1 and 10.6 ppm-m for our devices. This is 4 to 21 times higher than the sensitivity specified by the manufacturer. A large part of the inferior precision is attributed to low-frequency drifts, whereas high-frequency changes in the concentration are often well captured, as the similarity of the small features between hours 25 and 27 in Fig. 1 demonstrates. Drifts and step changes in the concentration occur up to 0.3 ppm (Fig. 1).
Most critical are changes in the concentration that can hardly be distinguished from fluctuations of the atmospheric concentrations. Some of the step changes are caused by activity related to the handling of the GF3 device (e.g. downloading data, checking time, checking reference cell quality etc.). It remains unclear though, what activity causes these step changes, since none of the activities is consistently causing such step changes. The internal calibrations of the GF3 seem to differ



between devices. Devices OP-1, OP-2 and OP-3 show systematically lower concentration measurements than the devices
OP-4 and OP-5. Application with paired devices need an intercalibration of the devices. However, it remains unclear to what
extent a side-by-side intercalibration can be transferred to the actual measurement setup, since relocation of the devices
might cause systematic changes, as indicated by the different regression coefficients for different intercomparison
campaigns.

**Author contribution**

TK designed and coordinated the field campaigns. MB performed the GasFinder3 measurements. CA provided the QCL
measurements. CH evaluated the data and prepared the manuscript with contributions from MB and AN. TK, AN and CA
reviewed the paper and gave constructive suggestions.

**Competing interests**

The authors declare that they have no conflict of interest.

**Acknowledgements**

Funding by the Swiss Federal Office for the Environment (Contracts 00.5082.P2I R254-0652; 06.0091.PZ tR281-0748;
10.0021.PJ / N253-1914) is gratefully acknowledged. We thank the operators of the wastewater treatment plants (WWTPs)
and of the biogas plant, the three farmers in the surroundings of the WWTPs and the Agroscope stations Tänikon and
Posieux for providing the sites for the measurements.

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



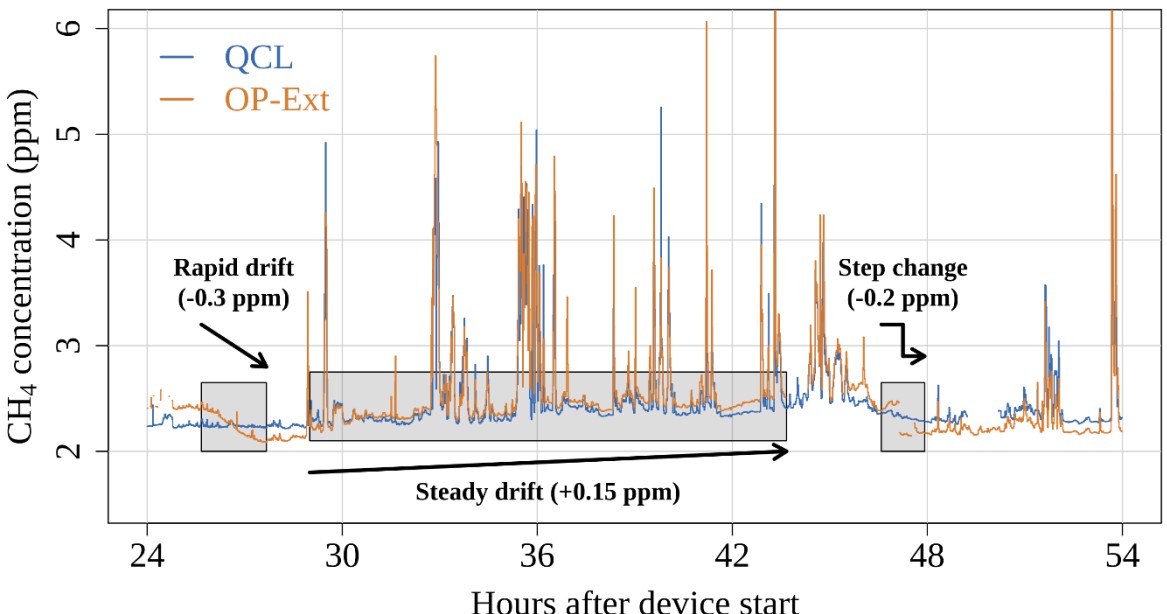

**Figure 1: Time series of the average CH₄ concentration (1-minute averages) measured with the QCL and the GF3 device OP-Ext during the intercomparison campaign P16. The figure shows a 30-hour window at the beginning of the campaign (1 to 2.5 days after instrument start). Three sub-periods with specific features are marked by grey shading.**





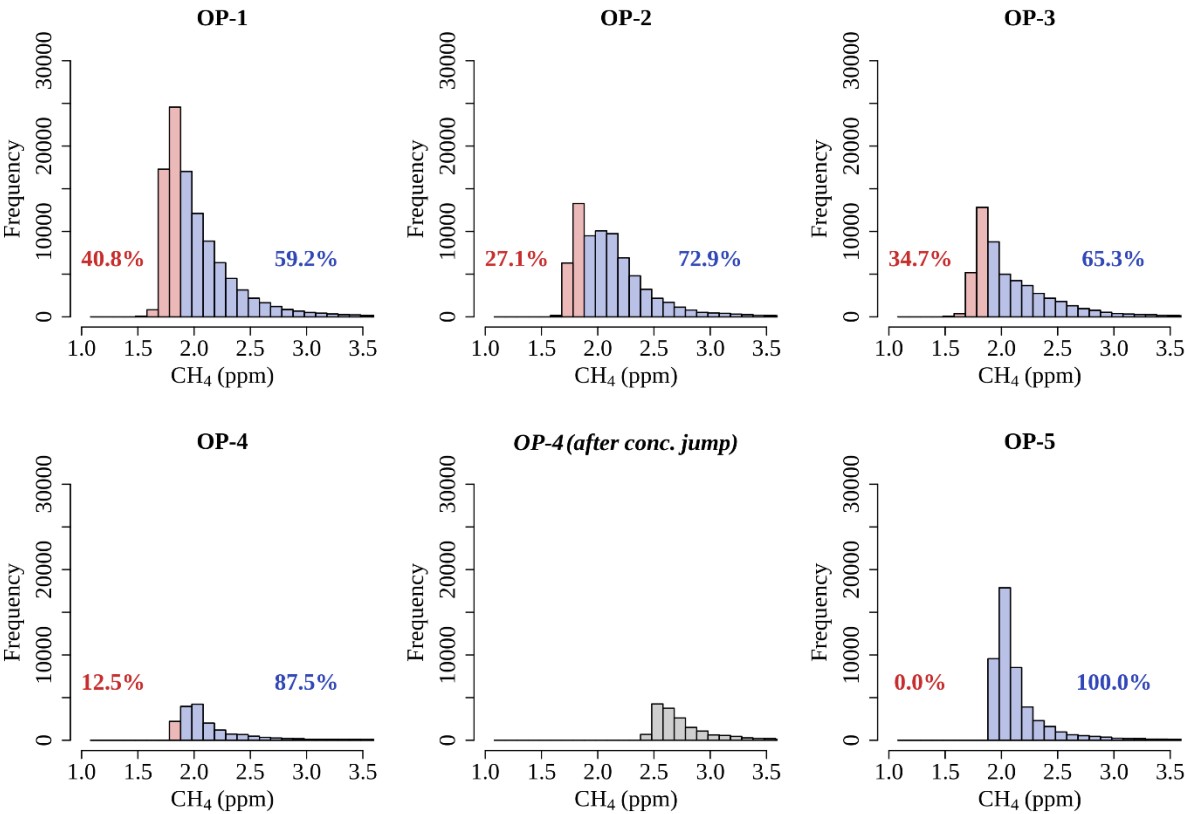

**Figure 2: Histograms of recorded 1-minute average concentrations of GF3 devices OP-1, OP-2, OP-3, OP-4 and OP-5. Few values**
**greater than 3.5 ppm are not shown. Blue: values > 1.88 ppm, red: values <= 1.88 ppm. Grey: Data from device OP-4 during the**
**campaign H19-3 that passed the quality check but has been omitted in the analysis due to an obvious jump in the concentration**
**(Fig. 3).**



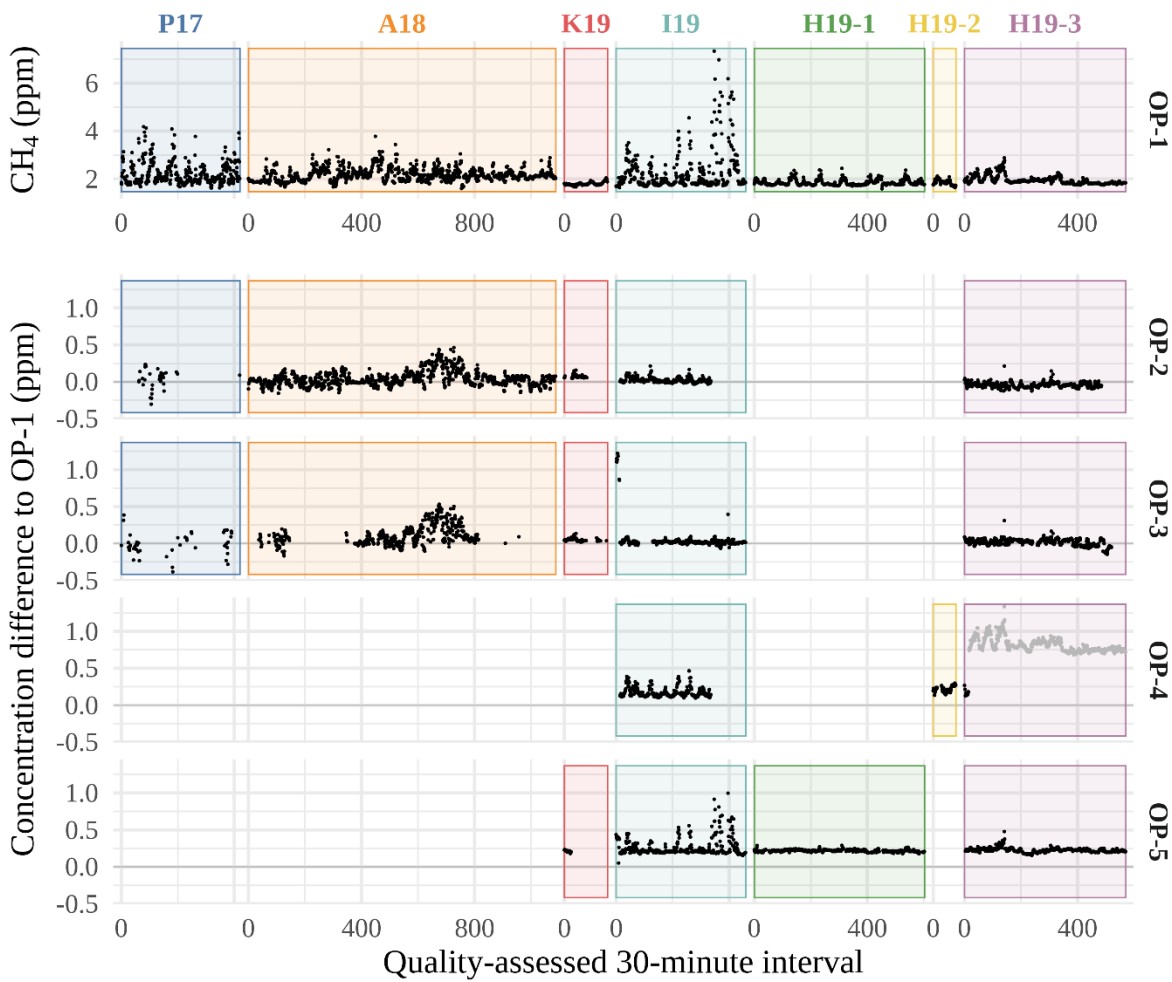

**Figure 3:** CH₄ concentrations recorded by OP-1 (30-minute averages) and the corresponding differences to OP-2, OP-3, OP-4 and
OP-5. Grey dots: Data from device OP-4 during the campaign H19-3 that passed the quality check but has been omitted in the
analysis due to an obvious jump in the concentration.



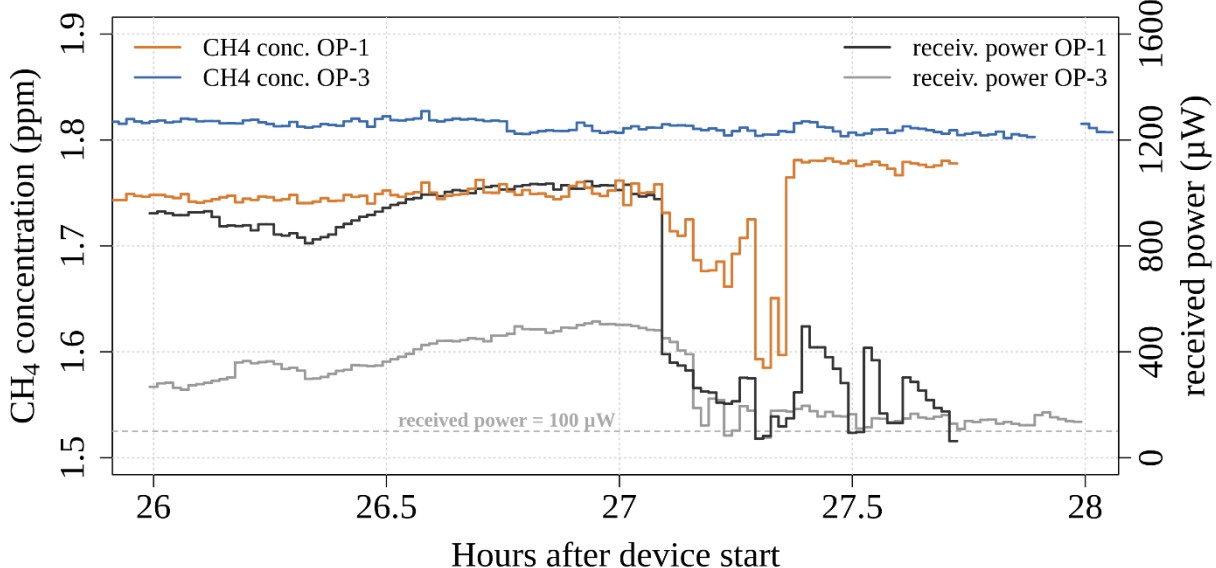

**Figure 4: Example of a concentration dent followed by a step change related to losses in the received power of device OP-1. The data was recorded during the intercomparison campaign K19 on 2019-04-26 between 2am and 4am. From hour 27 onwards, the data exhibit $R^2$ values above 0.98.**


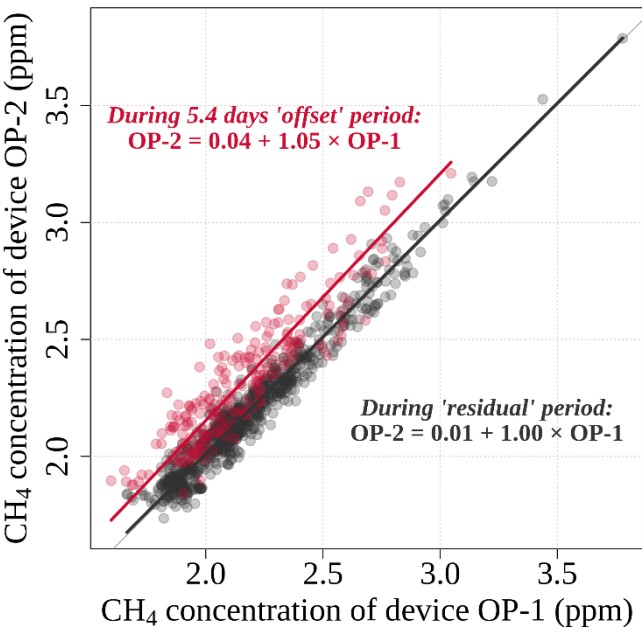

**Figure 5: Scatter plot of 30-minute data from OP-1 and OP-2 recorded during campaign A18. Deming regression lines and corresponding regression equations are shown for the 'offset' period and the remaining ('residual') period.**



**Table 1: GasFinder3-OP devices and their deployment in the different intercomparison campaigns. Details on the intercomparison campaigns are given in Table 2.**

| Name used in this study | Unit number | Year of manufacture | Intercomparison campaign | | | | | | | |
| --- | --- | --- | --- | --- | --- | --- | --- | --- | --- | --- |
| | | | P16 | P17 | A18 | K19 | I19 | H19-1 | H19-2 | H19-3 |
| **OP-Ext**[a] | CH4OP-30015 | 2016 | • | | | | | | | |
| **OP-1** | CH4OP-30017 | 2016 | | • | • | • | • | • | • | • |
| **OP-2** | CH4OP-30016 | 2016 | | • | • | • | • | | | • |
| **OP-3** | CH4OP-30018 | 2016 | | • | • | • | • | | | • |
| **OP-4** | CH4OP-30025 | 2019 | | | | | • | | • | • |
| **OP-5** | CH4OP-30026 | 2019 | | | | • | • | • | | • |

[a] on loan from Boreal Laser Inc.

**Table 2: Characteristics of the intercomparison campaigns (Camp.). Dur.: duration of the campaign. Air temperature: average (and minimum, maximum) values. Air press.: average air pressure.**

| Camp. | Location | Date | Dur. (days) | Instruments | Air temperature (°C) | Air press. (hPa) |
| --- | --- | --- | --- | --- | --- | --- |
| P16 | Posieux | 12 Oct – 01 Nov 2016 | 19.7 | QCL, 1 × GF3 | 7.5 ( -0.1 to 16.8) | 946 |
| P17 | Posieux | 19 Jul – 15 Aug 2017 | 26.8 | QCL, 3 × GF3 | 18.3 ( *7.3 to 32.2*) | 943 |
| A18 | Aadorf | 23 Oct – 21 Nov 2018 | 28.6 | 3 × GF3 | 6.3 ( *-2.4 to 17.9*) | 952 |
| K19 | Kaufdorf | 25 Apr – 30 Apr 2019 | 4.7 | 4 × GF3 | 7.7 ( *2.3 to 21.7*) | 955 |
| I19 | Ittigen | 19 Jul – 29 Jul 2019 | 10.2 | 5 × GF3 | 22.6 (*13.6 to 35.4*) | 951 |
| H19-1 | Hindelbank | 23 Sep – 07 Oct 2019 | 12.7 | 2 × GF3 | 13.9 ( *3.6 to 24.7*) | 956 |
| H19-2 | Hindelbank | 07 Oct – 14 Oct 2019 | 5.1 | 2 × GF3 | 12.7 ( *5.1 to 22.4*) | 959 |
| H19-3 | Hindelbank | 25 Oct – 06 Nov 2019 | 12.3 | 5 × GF3 | 9.7 ( *4.2 to 17.7*) | 953 |





**Table 3: Direct comparison of GF3 to QCL (30-minute averages) during campaigns P16 and P17. N: Number of 30-minute intervals. Path: Path length of GF3 device. Median C: Median concentration of the GF3 device. Rel. bias: Estimate of the GF3 relative bias. Precision: Estimate of the GF3 precision.**

| Campaign | Device | N | Path (m) | Median C (ppm) | Rel. bias (%) | Precision (ppm-m) |
|---|---|---|---|---|---|---|
| P16 | OP-Ext | 505 | 37 | 2.27 | -2.7 | 10.6 |
| P17 | OP-1 | 405 | 12 | 2.04 | -5.1 | 2.8 |
| P17 | OP-2 | 105 | 12 | 2.14 | -3.2 | 2.1 |
| P17 | OP-3 | 66 | 12 | 1.97 | -8.3 | 2.6 |

**Table 4: Direct comparison of GF3 devices OP-2 to OP-5 to the reference device OP-1 (30-minute averages). N: Number of 30-minute intervals. Path OP-1/OP-x: Path length of GF3 devices. Median C: Median concentration of OP-x. Rel. bias: Estimate of the GF3 relative bias. Precision: Estimate of the GF3 precision.**

| Campaign | Device (OP-x) | N | Path OP-1 (m) | Path OP-x (m) | Median C (ppm) | Rel. bias (%) | Precision (ppm-m) |
|---|---|---|---|---|---|---|---|
| P17 | OP-2 | 35 | 12 | 12 | 2.30 | 2.0 | 2.6 |
|  | OP-3 | 48 | 12 | 12 | 2.10 | -0.8 | 3.0 |
| A18 | OP-2 | 1081 | 37 | 37 | 2.15 | 0.9 | 5.5 |
|  | OP-3 | 465 | 37 | 37 | 2.24 | 2.6 | 8.8 |
| K19 | OP-2 | 53 | 170 | 118 | 1.83 | 2.7 | 3.6 |
|  | OP-3 | 82 | 170 | 176 | 1.82 | 1.8 | 6.1 |
|  | OP-5 | 25 | 170 | 118 | 1.98 | 8.0 | 2.7 |
| I19 | OP-2 | 322 | 110 | 110 | 1.89 | 0.6 | 5.3 |
|  | OP-3 | 404 | 110 | 110 | 1.96 | 0.6 | 3.4 |
|  | OP-4 | 317 | 110 | 110 | 2.03 | 5.4 | 4.9 |
|  | OP-5 | 456 | 110 | 110 | 2.10 | 7.3 | 5.3 |
| H19-1 | OP-5 | 542 | 112 | 111 | 2.01 | 7.9 | 4.0 |
| H19-2 | OP-4 | 66 | 65 | 65 | 2.04 | 7.5 | 5.9 |
| H19-3 | OP-2 | 483 | 110 | 50 | 1.86 | -1.7 | 5.2 |
|  | OP-3 | 485 | 110 | 51 | 1.93 | 0.9 | 6.7 |
|  | OP-5 | 559 | 110 | 109 | 2.11 | 7.7 | 5.5 |





**Table 5: Coefficients from the Deming regression between OP-1 and OP-2 to OP-5 with 30-minute averaged data. Standard errors of the estimates are given in parentheses. Only campaigns were analyzed, where N > 20 and the concentration range was large enough (difference between 0.025 and 0.975 quantiles greater than 0.4 ppm). Dev.: GF3 device used as regressand. Cmp.: Intercomparison campaign. N: Number of 30-minute intervals. $\sigma_{resid}$: Standard deviation of the model residuals. $\Delta C_{yppm}$: Predicted difference between the OP-x concentration and the OP-1 concentration at a level of y ppm (2 ppm or 4 ppm). Lower and upper bounds of the 95 % confidence interval are given in parentheses. For each device and concentration level, intercomparison campaigns not sharing a super positioned letter exhibit significantly different $\Delta C$.**

| Dev. | Cmp. | N | Intercept (ppm) | Slope (-) | $\sigma_{resid}$ (ppm) | $\Delta C_{2ppm}$ (ppm) | $\Delta C_{4ppm}$ (ppm) |
|------|------|------|-----------------|-----------|------------------------|-------------------------|-------------------------|
| OP-2 | P17 | 35 | 0.15 (0.11) | 0.96 (0.05) | 0.09 | 0.06[ab] (-0.13, 0.24) | -0.03[ab] (-0.28, 0.22) |
|      | A18 | 1081 | -0.04 (0.03) | 1.04 (0.01) | 0.07 | 0.04[ab] (-0.10, 0.17) | 0.11[ab] (-0.04, 0.25) |
|      | I19 | 322 | -0.10 (0.01) | 1.06 (0.00) | 0.02 | 0.02[a] (-0.01, 0.05) | 0.14[a] (0.10, 0.17) |
|      | H19-3 | 483 | -0.12 (0.03) | 1.04 (0.02) | 0.02 | -0.04[b] (-0.09, 0.01) | 0.04[b] (-0.05, 0.12) |
| OP-3 | P17 | 48 | -0.01 (0.11) | 1.00 (0.05) | 0.11 | -0.01[a] (-0.23, 0.21) | -0.02[a] (-0.32, 0.29) |
|      | A18 | 465 | -0.09 (0.06) | 1.10 (0.03) | 0.09 | 0.10[a] (-0.09, 0.28) | 0.29[a] (0.07, 0.50) |
|      | I19 | 404 | 0.03 (0.01) | 1.01 (0.01) | 0.11 | 0.04[a] (-0.19, 0.27) | 0.05[a] (-0.18, 0.28) |
|      | H19-3 | 485 | -0.14 (0.04) | 1.08 (0.02) | 0.03 | 0.02[a] (-0.04, 0.08) | 0.18[a] (0.09, 0.28) |
| OP-4 | I19 | 317 | -0.12 (0.01) | 1.14 (0.00) | 0.01 | 0.16 (0.14, 0.19) | 0.44 (0.41, 0.47) |
| OP-5 | I19 | 456 | -0.03 (0.01) | 1.13 (0.00) | 0.03 | 0.22[a] (0.16, 0.28) | 0.47[a] (0.41, 0.53) |
|      | H19-1 | 542 | 0.14 (0.01) | 1.04 (0.01) | 0.01 | 0.22[a] (0.20, 0.24) | 0.31[b] (0.27, 0.35) |
|      | H19-3 | 559 | 0.03 (0.02) | 1.10 (0.01) | 0.02 | 0.23[a] (0.20, 0.26) | 0.43[a] (0.37, 0.49) |