# Peer review of "Performance of open-path GasFinder3 devices for CH4 concentration measurements close to ambient levels"

_Atmospheric Measurement Techniques, 2020_

## Referee Comment (RC1) · Anonymous Referee #1 · 28 Oct 2020

Review of "Performance of open-path GasFinder3 devices for CH4 concentration measurements close to ambient levels" by Hani et al., submitted to Atmospheric Measurement Techniques (amt-2020-236). This paper describes field tests of Boreal Laser's open-path CH4 measurement system, focusing on several performance metrics. Boreal Laser's GasFinder system has been an important gas sensor for over 20 years. With its relatively low price, extreme robustness, and simplicity of operation, it continues to play an important role in fenceline gas monitoring and emission source rate quantification studies. The subject of this paper is appropriate for the AMT journal, and it will be of interest to a substantial audience. The manuscript is well written and the analysis and conclusions seem sound. I have made mostly minor comments on this

work. My recommendation is to accept the manuscript for publication.

Major Comment

1. The evidence of poor GasFinder performance (compared to the manufacturer's specifications) is convincing. Two of the analyses in this study are most important. First is the accuracy and precision estimates derived from the QCL comparisons. This gives a good estimate of the performance of an "off-the-shelf" GasFinder. I do suggest a refinement to this analysis. Can the authors cross-calibrate the lasers and the QCL (i.e., force agreement in the long-term average concentration), and then recalculate the precision? This situation would be the best-case scenario for a laser application. The second interesting analysis shows the variability of the laser cross-calibrations with time. This is perhaps the most important practical finding, as in the past users accepted poor agreement between lasers, believing that a cross-calibration can eliminate or reduce that problem. The findings from this study show that is not the case.

2. Can the authors make a case that in some circumstances a GasFinder based IDM measurement (upwind & downwind lasers) can provide a reasonably accurate determination of emissions (e.g., < 20% error)? For example, large cattle feedlots can have a CH4 concentration rise (above ambient) in the feedlot interior of approximately 1-2 ppm. Based on the errors given in Table 5, are there upwind-downwind laser combinations that might give an emission rate calculation within 20% of the true rate? Such an exercise would be insightful for GasFinder users, and provide for some added perspective.

Minor Comments

3. Line 11: The Boreal Laser company should be identified with the first reference to the GasFinder.

4. Line 12 & 13: Do the authors need to tie this work to agricultural emissions? GasFinders are used more broadly than this (I am aware of their use in CH4 measurements

at mines, at heavy industries, waste-water treatment plants, etc.). In terms of the entire manuscript, one could delete almost every instance of "agricultural" from the paper.

5. Line 16: "We investigated the uncertainty of six GF3 devices from side by side intercomparison measurements and comparisons to a closed-path quantum cascade laser device". It is important to add that the comparison was made at near-ambient levels of CH4 (and indicating the concentration range, e.g., 1.8 – 2.4 ppm).

6. Line 29: "It is in common to many IDM applications that the concentration enhancement related to agricultural CH4 sources is small, typically between 0.05 and 0.5 ppm." This "problem" is not unique to agricultural sources, so the "agricultural" qualifier is unneeded.

7. Line 32: "They are based e.g. on the determination of the absorption over a small wavelength range e.g. in the infrared spectrum (tunable diode laser technique for CH4; DeBruyn et al., 2020)." Awkward and unclear sentence. Rewrite.

8. Line 37: "On the other hand, it is more difficult to assess and control the quality of measurements by open-path gas analyzers in comparison to closed-path instruments." Very good point.

9. Line 45: "In this paper, we focus on the GasFinder3-OP (GF3) system for CH4 measurements (Boral Laser Inc, Edmonton Canada) with the 'Lo-Range' calibration option." Some explanation for the "Lo-Range" option is needed. Is this a specific type of laser? Does it use a different fitting curve in the concentration calculation? But I would say this is an unneeded detail in the broad objectives paragraph. Also, correct the company name to "Boreal".

10. Line 62: "The output data in units of ppm-m was converted to the path-averaged concentration C in units of ppm (i.e. divided by the single path length) and corrected with temperature and pressure . . ." Use "one-way" pathlength rather than "single".

11. Line 70: "According to the manufacturer, a valid concentration measurement can

be expected if the 'received power' of the reflected incoming laser beam is in the range of 50 to 3000 $\mu$W . . ." Is power a routine output variable from the GasFinder?

12. Line 78:" Two campaigns, P16 and P17, with a focus on the comparison . . . close to an animal housing facility (approx. 100 m north)." Does the sensor proximity to the animal housing mean the CH4 concentrations were elevated over ambient levels? Other campaigns also took place near gas sources. The authors might want to clarify whether they are looking at true ambient concentrations, or concentrations that ranged from ambient to somewhat above ambient, or near-ambient, etc.

13. Line 220: "However, it remains unclear to what extent a side-by-side intercalibration can be transferred to the actual measurement setup, since relocation of the devices might cause systematic changes, as indicated by the different regression coefficients for different intercomparison campaigns". Excellent and very important point.

---

## Referee Comment (RC2) · Anonymous Referee #2 · 16 Nov 2020

This manuscript focuses on the evaluation of biases and precision in GasFinder GF3 open path methane sensors, a commercially available sensor currently utilized by multiple organizations for emission monitoring. Multiple GF3 units were compared over multiple field intensives by comparison with both each other and an in situ analyzer. The manuscript represents a substantial scientific contribution that is within the scope of AMT and utilizes valid scientific approaches and methods. I recommend that the manuscript should be accepted following address of the below minor issues, primarily concerning the justification for the statistical methods used.

• Abstract, line 18: precision at 1 sigma?

[Figure]

• Section 2.1, Line 69: even though details are described in reference, there should be a brief further description as to how concentrations are calculated and how the calibration waveform is measured. Otherwise, it is more difficult to understand the metrics discussed in this paragraph.

• Section 2.2, Line 80: how was the QCL instrument calibrated? How often? What scale was the calibrant traceable to (e.g. WMO)?

• Section 2.3, paragraph 2: I do not feel there was sufficient justification for the use of median based statistics over Gaussian, especially when the result was to use Gaussian assumptions to convert the median statistics to Gaussian ones. There should at least be a discussion as to why the outliers are expected to be as prevalent in a non-Gaussian manner as to justify this approach.

• Section 2.3, line 120: I don't understand the propagation justification to add the sqrt(2) factor. It seems to me that there are some math steps or justification missing to explain how the error is being propagated.

---

## Author Comment (AC1) · 30 Dec 2020

We thank for the valuable commenting of Referee #1 and the opportunity to revise our manuscript. We fully addressed the reviewer's comments as described below. The column 'Line' is referring to the line number in the revised manuscript.

| Comments Referee #1 | Authors response | Line |
|---|---|---|
| 1. The evidence of poor GasFinder performance (compared to the manufacturer's specifications) is convincing. Two of the analyses in this study are most important. First is the accuracy and precision estimates derived from the QCL comparisons. This gives a good estimate of the performance of an "off-the-shelf" GasFinder. I do suggest a refinement to this analysis. Can the authors cross-calibrate the lasers and the QCL (i.e., force agreement in the long-term average concentration), and then recalculate the precision? This situation would be the best-case scenario for a laser application. The second interesting analysis shows the variability of the laser cross-calibrations with time. This is perhaps the most important practical finding, as in the past users accepted poor agreement between lasers, believing that a cross-calibration can eliminate or reduce that problem. The findings from this study show that is not the case. | We addressed the issue (long-term forced agreement between sensors) by separating the total uncertainty into the average systematic bias and the precision (=variability of short-term bias/difference) in our analysis. The presented precision is marginally affected by the instruments long-term calibration. An additional adjustment of the long-term span (which is, in our case, underestimated by the factory calibration) would even slightly worsen the precision estimates (+2% to +7% increase in the estimated values). Thus, using the original factory calibration results in a more optimistic (i.e. smaller) value of the GF precision. | |
| 2. Can the authors make a case that in some circumstances a GasFinder based IDM measurement (upwind & downwind lasers) can provide a reasonably accurate determination of emissions (e.g., < 20% error)? For example, large cattle feedlots can have a CH4 concentration rise (above ambient) in the feedlot interior of approximately 1-2 ppm. Based on the errors given in Table 5, are there upwind-downwind laser combinations that might give an emission rate calculation within 20% of the true rate? Such an exercise would be insightful for GasFinder users, and provide for some added perspective. | It is very difficult to a give general estimation or recommendation concerning the GF3 induced error for IDM measurements. This is because the concentration difference of IDM applications can vary (over an order of magnitude or more) depending on the source strength, the geometry of the experimental setup, and the turbulence conditions. The estimated precision of 2.1 to 10.6 ppm-m corresponds to an uncertainty in the concentration difference of 0.06 to 0.30 ppm for a path length of 50 m and 0.01 to 0.06 ppm for a path length of 250 m, given that the systematic bias has been eliminated by inter-calibration and given that the instruments' span has not been altered and, further, given that the GF3 don't exhibit drift and offset features as discussed in Section 3.1. Taking the mentioned example of a large cattle feedlot with 1 ppm concentration rise and a path length of 100 m, this would result in an integrated (one-way) concentration of 100 ppm-m and, therefore, to an uncertainty of roughly 2% to 10% in the concentration measurement, which would suggest a reasonably accurate application of the GF3 measurements in IDM. In contrast, for a typical farm in Switzerland with | |

| | only 50 dairy cows, the measured concentration difference can be much smaller. Moreover, it must be considered, that the estimated instrument precision is valid for half-hourly concentration averages and estimating the average emission from a long-term measurement series can significantly reduce the uncertainty in the final emission estimate (Bühler et al, submitted). Therefore, we prefer not to give specific calculation examples for IDM application errors in the manuscript, because they easily can be misinterpreted. | |
|---|---|---|
| 3. Line 11: The Boreal Laser company should be identified with the first reference to the GasFinder. | The sentence was changed to:

"Open-path measurements of methane ($CH_4$) with the use of GasFinder systems (Boreal Laser Inc, Edmonton Canada) has been frequently used for emission estimation with the inverse dispersion method (IDM), particularly from agricultural sources." | Lines 11 to 13 |
| 4. Line 12 & 13: Do the authors need to tie this work to agricultural emissions? GasFinders are used more broadly than this (I am aware of their use in CH4 measurements at mines, at heavy industries, waste-water treatment plants, etc.). In terms of the entire manuscript, one could delete almost every instance of "agricultural" from the paper. | The authors were mainly aware of papers discussing IDM uses for estimating agricultural emissions. However, Referee #1 notes correctly that there is no need to tie this work to agricultural emissions.

The sentence was changed to:

"Open-path measurements of methane ($CH_4$) with the use of GasFinder systems (Boreal Laser Inc, Edmonton Canada) has been frequently used for emission estimation with the inverse dispersion method (IDM), particularly from agricultural sources. It is common to many IDM applications that the concentration enhancement related to $CH_4$ sources is small, typically between 0.05 and 0.5 ppm, and accurate measurements of $CH_4$ concentrations are needed at concentrations close to ambient levels." | Lines 11 to 15 |
| 5. Line 16: "We investigated the uncertainty of six GF3 devices from side by side intercomparison measurements and comparisons to a closed-path quantum cascade laser device". It is important to add that the comparison was made at near-ambient levels of CH4 (and indicating the concentration range, e.g., 1.8 – 2.4 ppm). | We added the following sentences:

"The comparisons were made at near-ambient levels of $CH_4$ (85 % of measurements below 2.5 ppm) with occasional phases of elevated concentrations (max. 8.3 ppm)." | Lines 17 to 19 |
| 6. Line 29: "It is in common to many IDM applications that the concentration enhancement related to agricultural CH4 sources is small, typically between 0.05 and 0.5 ppm." This "problem" is not unique to agricultural | The term "agricultural" was removed from the sentence:

"It is in common to many IDM applications that the concentration enhancement related to $CH_4$ sources is small, typically between 0.05 and 0.5 ppm." | Lines 30 to 32 |

| | | |
|---|---|---|
| sources, so the "agricultural" qualifier is unneeded. | | |
| 7. Line 32: "They are based e.g. on the determination of the absorption over a small wavelength range e.g. in the infrared spectrum (tunable diode laser technique for CH4; DeBruyn et al., 2020)." Awkward and unclear sentence. Rewrite. | The sentence was deleted, and the preceding sentence was changed to:

"In recent years, optical open-path instruments became commercially available that determine the path-integrated $CH_4$ concentration over measurement path lengths of up to several 100 meters." | Lines 33 to 34 |
| 8. Line 37: "On the other hand, it is more difficult to assess and control the quality of measurements by open-path gas analyzers in comparison to closed-path instruments." Very good point. | We thank the reviewer for the supporting appraisal. | |
| 9. Line 45: "In this paper, we focus on the GasFinder3-OP (GF3) system for CH4 measurements (Boral Laser Inc, Edmonton Canada) with the 'Lo-Range' calibration option." Some explanation for the "Lo-Range" option is needed. Is this a specific type of laser? Does it use a different fitting curve in the concentration calculation? But I would say this is an unneeded detail in the broad objectives paragraph. Also, correct the company name to "Boreal". | The sentence was changed to:

"In this paper, we focus on the GasFinder3-OP (GF3) system for $CH_4$ measurements (Boreal Laser Inc, Edmonton Canada) with the "Lo-Range" methane option (i.e. factory calibrated for a detection range between 2 and 8500 ppm-m)." | Lines 45 to 46 |
| 10. Line 62: "The output data in units of ppm-m was converted to the path-averaged concentration C in units of ppm (i.e. divided by the single path length) and corrected with temperature and pressure …" Use "one-way" pathlength rather than "single". | We changed 'single path length' to 'one-way path length'. | Lines 60, 62 and 128 |
| 11. Line 70: "According to the manufacturer, a valid concentration measurement can be expected if the 'received power' of the reflected incoming laser beam is in the range of 50 to 3000 µW …" Is power a routine output variable from the GasFinder? | The sentence was extended to:

"Together with the concentration measurement, the supporting parameters 'received power' (of the reflected incoming beam) and 'R2' (the goodness of fit between the sample and the calibration waveform) are provided as standard outputs of the GF3 instruments. According to the manufacturer, a valid concentration measurement can be expected if the following constraints are met: 'received power' is in the range of 50 to 3000 µW and 'R2' is above 0.85 (Boreal Laser Inc., 2018b)." | Lines 69 to 72 |
| 12. Line 78:" Two campaigns, P16 and P17, with a focus on the comparison … close to an animal housing facility (approx. 100 m north)." Does the sensor proximity to the animal housing mean the CH4 concentrations were elevated over ambient levels? Other campaigns also took place near gas sources. The authors might want to clarify whether they are looking at true ambient | We changed the previous sentence to:

"In total, eight intercomparison campaigns were conducted at different sites in Switzerland with varying ranges of near-ambient concentrations of $CH_4$ (Table 2)." | Lines 78 to 79

Table 2 |

| | | |
|---|---|---|
| concentrations, or concentrations that ranged from ambient to somewhat above ambient, or near-ambient, etc. | Moreover, we added a summary of the measured concentration (average, minimum and maximum) for each campaign to Table 2. | |
| 13. Line 220: "However, it remains unclear to what extent a side-by-side intercalibration can be transferred to the actual measurement setup, since relocation of the devices might cause systematic changes, as indicated by the different regression coefficients for different intercomparison campaigns". Excellent and very important point. | We appreciate this positive feedback. | |

References:

Bühler M., Häni C., Ammann C., Mohn J., Neftel A., Schrade S., Zähner M., Zeyer K., Brönnimann S., and Kupper T.: Assessment of the inverse dispersion method for the determination of methane emissions from a dairy housing, submitted to Agric. For. Meteorol.

---

## Author Comment (AC2) · 30 Dec 2020

We thank for the valuable commenting of Referee #2 and the opportunity to revise our manuscript. We fully addressed the reviewer's comments as described below. The column 'Line' is referring to the line number in the revised manuscript.

| Comments Referee #2 | Authors response | Line |
|---|---|---|
| 1. Abstract, line 18: precision at 1 sigma? | We clarified the sentence to:

"...and a precision for half-hourly data between 2.1 and 10.6 ppm-m (half width of the 95 % confidence interval) was estimated." | 19 to 20 |
| 2. Section 2.1, Line 69: even though details are described in reference, there should be a brief further description as to how concentrations are calculated and how the calibration waveform is measured. Otherwise, it is more difficult to understand the metrics discussed in this paragraph. | The manufacturer does not provide detailed information on the derivation of the concentration.
The manufacturer states that the calibration waveform is fitted to the measured waveform with the use of the linear least-squares regression analysis (Appendix F, Boreal Laser Inc., 2018b).
However, we don't have more details on the employed fitting procedure and the measurement of the calibration waveform.

We changed the sentence to:

"Together with the concentration measurement, the supporting parameters 'received power' (of the reflected incoming beam) and 'R2' (the goodness of fit between the sample and the calibration waveform) are provided as standard outputs of the GF3 instruments. According to the manufacturer, a valid concentration measurement can be expected if the following constraints are met: 'received power' is in the range of 50 to 3000 µW and 'R2' is above 0.85 (Boreal Laser Inc., 2018b)." | 69 to 72 |
| 3. Section 2.2, Line 80: how was the QCL instrument calibrated? How often? What scale was the calibrant traceable to (e.g. WMO)? | The used QCL instrument provides absolute concentration measurements (based on absorption spectra from the HITRAN database) without the need for empirical calibration (Nelson et al., 2004). Nevertheless, the instrument was tested occasionally using cylinder standards of 1.50 ppm and 2.00 ppm $CH_4$ (with a factory certified accuracy of 2%). It generally agreed with the standards within their uncertainty range. | |
| 4. Section 2.3, paragraph 2: I do not feel there was sufficient justification for the use of median based statistics over Gaussian, especially when the result was to use Gaussian assumptions to convert the median statistics to Gaussian ones. There should at least be a discussion as to why the outliers are expected to be as prevalent in a non-Gaussian manner as to justify this approach. | As mentioned in the text, we have chosen the median based statistics, because it is less sensitive to outliers and to deviations from an ideal Gaussian error distribution.

In fact, there's only a marginal difference in precision estimation between GF3 and QCL, when using Gaussian statistics. However, it has a substantial impact on estimates (mainly) from | 115 to 116 |

| | | |
|---|---|---|
| | two campaigns comparing GF3 side-by-side measurements, where the distribution of ∆C clearly differed from an ideal Gaussian distribution (see Fig. 3) and thus the precision estimate based on Gaussian statistic would be clearly influenced by a few large values.

For clarification, we enhanced the text as follows:

"The ∆C data partly showed significant deviations (asymmetry, outliers) from an ideal Gaussian distribution. Thus, for analyzing the difference between devices, the median ..." | |
| 5. Section 2.3, line 120: I don't understand the propagation justification to add the sqrt(2) factor. It seems to me that there are some math steps or justification missing to explain how the error is being propagated. | As mentioned in the second sentence preceding Eqs. 1 and 2, the sqrt(2) factor was introduced to partition the uncertainty (precision) of the concentration difference ∆C to the two individual concentration measurements that are assumed to be of equal magnitude.

For clarification we slightly modified the text as follows:

"The estimates of bias and precision for ∆C can be partitioned equally to the concentrations of both intercompared devices by dividing by the square root of 2 (according to Gaussian error propagation)." | 121 to 123 |

References:

Boreal Laser Inc.: GasFinder3-OP Operation Manual, Part No. NDC-200036, 2018b.

Nelson, D. D., McManus, B., Urbanski, S., Herndon, S., and Zahniser, M. S.: High precision measurements of atmospheric nitrous oxide and methane using thermoelectrically cooled mid-infrared quantum cascade lasers and detectors, Spectrochimica acta. Part A, Molecular and biomolecular spectroscopy, 60, 3325–3335, doi:10.1016/j.saa.2004.01.033, 2004.